# Study on the Restoration of Class II Carious Cavities by Virtual Methods: Simulation of Mechanical Behavior

**DOI:** 10.3390/jfb14070354

**Published:** 2023-07-05

**Authors:** Mihaela Jana Țuculină, Adela Nicoleta Staicu, Maria Cristina Munteanu, Cristian Niky Cumpătă, Bogdan Dimitriu, Ana Maria Rîcă, Maria Cristina Beznă, Dragoș Laurențiu Popa, Alexandru Dan Popescu, Tiberiu Țîrcă

**Affiliations:** 1Department of Endodontics, Faculty of Dentistry, University of Medicine and Pharmacy of Craiova, 200349 Craiova, Romania; mtuculina@yahoo.com (M.J.Ț.); r_ana_maria22@yahoo.com (A.M.R.); alexandrudanpopescu20@gmail.com (A.D.P.); 2Department of Oral and Maxillofacial Surgery, Faculty of Dental Medicine, University of Medicine and Pharmacy of Craiova, 200349 Craiova, Romania; cristina_omf@yahoo.com; 3Faculty of Dental Medicine, University Titu Maiorescu of Bucharest, 67A Gheorghe Petrascu Str., 031593 Bucharest, Romania; nikycumpata@yahoo.com; 4Department of Endodontics, Faculty of Dentistry, University of Medicine and Pharmacy Carol Davila Bucharest, 050474 Bucharest, Romania; bogdan.dimitriu@umcfcd.ro; 5Department of Pathophysiology, Faculty of Dentistry, University of Medicine and Pharmacy of Craiova, 200349 Craiova, Romania; bezna.mariacristina@gmail.com; 6Department of Automotive, Transportation and Industrial Engineering, Faculty of Mechanics, University of Craiova, 200478 Craiova, Romania; 7Department of Oro-Dental Prevention, Faculty of Dental Medicine, University of Medicine and Pharmacy of Craiova, 200349 Craiova, Romania; tiberiu.tirca@yahoo.com

**Keywords:** Bichacho, bruxism, class II cavity, dental restoration, fluid composites, injection molding, paste composites, snow plow

## Abstract

The restoration of class II cavities is predominantly carried out with composite materials. Due to the high failure rate in restoring this type of cavity, composite materials with much-improved properties and new application techniques have been promoted. The study aimed to analyze the mechanical behavior of several topical composite materials (nanocomposites, nanohybrids and ormocer) using different application techniques. In a lower second molar, a class II occlusal cavity was prepared. As filling materials, we used the following combinations: Admira Fusion and Admira Fusion Flow, Grandio and Grandio Flow, Filtek Supreme XT and Filtek Supreme Flow. These were applied using a snow plow, injection molded and Bichacho techniques. Three-dimensional scanning of the molar with the prepared cavity was performed, and then scanning of each layer of added composite material was performed, obtaining three-dimensional models. The virtual molar models were analyzed with software specific to the finite element analysis method, where their physical-mechanical properties were entered and assigned to the components of the virtual molar. Simulations at high forces specific to bruxism were then carried out and analyzed, and compared. The values of displacements and strain, for all six analyzed situations, are relatively small (range from 5.25 × 10^−6^–3.21 × 10^−5^ for displacement, 6.22 × 10^−3^–4.34 × 10^−3^ for strain), which validates all three methods and the materials used. As far as the stress values are concerned, they are similar for all methods (250–300 MPa), except for the snow plow and injection-molded techniques using Grandio and Grandio Flow composites, where the maximum von Mises stress value was more than double (approximately 700 MPa). When using the combination of Grandio and Grandio Flow materials, the 1 mm thickness of the fluid composite layer was found to have a major influence on occlusal forces damping as opposed to 0.5 mm. Therefore, the Bichacho technique is indicated at the expense of the snow plow and injection-molded techniques. The composite materials used by us in this study are state-of-the-art, with clear indications for restoring cavities resulting from the treatment of carious lesions. However, their association and application technique in the case of Class II cavities is of clinical importance for resistance to masticatory forces.

## 1. Introduction

Traditionally, amalgam has been used for filling the cavities of lateral teeth, but with the improvement in the properties of composite resins, they have increasingly gained ground in the direct restoration of posterior teeth [1]. Although proximal caries are second in frequency after occlusal ones, restoring class 2 cavities is much more difficult to achieve [2]. The restoration of second-class cavities using composite materials has a number of disadvantages related to the difficulty of technique application and the inherent polymerization contraction properties of these materials [3]. Visibility and difficult access determine the risk of not achieving a good marginal closure and, at the same time, through the loss of the marginal ridge, the greater risk of fracture at high occlusal demands [4]. Studies have shown that complex Class II restorations are more prone to clinical complications and have shorter longevity [5,6]. The use of incorrect techniques in restoring class II cavities can lead to postoperative sensitivity, marginal microleakage with the appearance of secondary caries and excessive wear [7]. Considering the complications listed above and taking into account the increasing frequency of bruxism among patients, bruxism currently represents a significant challenge in restorative dentistry because extreme forces develop in this condition, which acts cyclically. Causes mechanical overloading of restorations in patients with bruxism leads to wear, fracture and failure in general [8].

Class II restorations are more prone to fracture due to the involvement of the marginal ridge, the high stresses occurring in the isthmus area, and the buckling effect causing horizontal stresses that weaken the cavity walls and lead to fractures [9,10]. Besides this, the long-term strength of occlusal-proximal restorations is also influenced by the quality of the used materials, the presence of cavity wall lining or patient-related factors such as bruxism [11]. Despite limited and inconclusive scientific evaluations [12,13], there is an increased use in the general practice of a flowable composite as a liner in cavities to relieve stresses occurring in Class I and II restorations [14,15]. Various authors have suggested the use of an elastic liner layer under posterior composite restorations, which acts as a stress-absorbing intermediate layer, thereby reducing polymerization shrinkage. Mc Lean and Wilson presented this technique in 1977 as the sandwich technique, in which resin-modified glass ionomer cement or fluid composites were used as liners on the cavity floor, cured and then followed by the addition of paste composite layers [16,17].

Nowadays, new techniques have appeared (snow plow and injection-molded), in which the two layers of different materials (paste and fluid) are light-cured simultaneously, leading to a decrease in both the internal stresses in the restoration and the number of clinical steps involved [18,19]. Given the existence to date of countless studies analyzing the efficiency of these techniques in achieving marginal closure [19,20,21] but insignificant in terms of the mechanical strength of fillings made by these techniques, it was set out to analyze the mechanical behavior of class II fillings using simulations using the finite element method. 

The choice of this method is justified by the fact that, since the teeth present different morphologies and individual variations in structure and content of organic and inorganic components, for reliable analysis, an appropriate 3D method was considered [22]. Unlike clinical and experimental research, the finite element method allows the determination of von Mises stress, displacements and deformations inside the tooth. The finite element method allows safe, fast and relatively low-cost simulation, proving its usefulness in numerous simulations in the medical field [23,24,25,26].

Simulations were performed on a second mandibular molar using several combinations of composite resins, applied through a centripetal restorative technique: snow plow, injection-molded and Bichacho techniques. Starting from the finding that composite materials with Young’s modulus similar to dentine can withstand pressures of 90–300 MPa, it was wanted to analyze if they can also withstand forces greater than 700 N, similar to those developed by patients with bruxism [27].

The aim of the study is to determine the mechanical behavior of some combinations of paste and flow composites, arranged in the cavity and photopolymerized differently, to the forces generated by high occlusal stress. 

The study′s null hypothesis (h0) was that all combinations of restorative materials behave identically when extreme occlusal forces are applied, regardless of the method of application and light curing. 

This study is significant because it will provide clinicians with evidence-based recommendations regarding the mechanical changes undergone by composite materials when subjected to high occlusal forces, such as those occurring in bruxism. Ultimately, the study may lead to improved clinical outcomes and a higher success rate for the restorative treatment of Class II cavities.

## 2. Materials and Methods

For the present study, a study model was used of a caries-free lower second molar without dental restorations, in which an occlusal-proximal cavity of medium depth was prepared, with the gingival wall located 1 mm above the enamel-cement junction. The occlusal-proximal cavity was restored by the centripetal method, i.e., by filling first the vertical cavity on the mesial side and then the horizontal cavity on the occlusal side.

In the case of the vertical component, three different techniques were used for the filling: the snow plow technique, the injection-molded technique, and the Bichacho technique (classic technique). In the snow plow technique, after the adhesive system has been applied and polymerized, a 0.5 mm layer of fluid composite is applied to the gingival wall, and the paste composite is applied on top of this unpolymerised layer. After condensation, the excess is removed, and the resulting layer is polymerized [27,28,29]. The injection-molded technique differs from the previous one in that the adhesive, the fluid composite and the paste composite are applied one at a time, but each layer is not light-cured separately, but all three layers simultaneously [23,24]. In the Bichacho technique, the polymerization of the used materials (adhesive, fluid composite and paste composite) is carried out separately. In the first phase, the adhesive is applied and photopolymerized. Over the adhesive layer, a 1 mm thick layer of fluid composite is applied and light-cured. On the last layer, the paste composite is applied, thus completing the restoration of the proximal wall and is light-cured [25].

Occlusal cavity restoration was achieved by applying a 0.5 mm thick layer of fluid composite for the injection-molded and snow plow technique and 1 mm for the Bichacho technique to maintain the proportions of the layers at the vertical wall level. After light-curing this layer, cuspid by cuspid was restored, applying oblique layers and recreating the occlusal morphology. Polymerization was performed bidirectionally for each layer applied [30]. In our study, the composite materials highlighted in Table 1 were used. Both flow and paste composites were used, considering the need for both types of composites for the correct application of the three restoration methods. Paste-flow combinations belonging to the same manufacturer were used, and three combinations of current materials were selected.

As an adhesive system G-Premio BOND, a one-component, 8th-generation universal adhesive, was used. However, in the finite element method simulations, the adhesive used for cavity restoration was no longer modeled three-dimensionally but was simulated by Bonded contacts.

In this study, the materials are considered to be homogeneous, isotropic and linearly elastic [31,32]. Considering that to perform the proposed simulations, the physical-mechanical properties of the composite materials used, namely density, Young’s modulus of elasticity, and Poisson’s ratio, were required. Specialty mentions in the literature mention the following values highlighted as shown in Table 2, which were used for the present study [33,34,35,36,37,38,39,40,41,42,43,44,45,46,47,48,49,50,51,52].

Due to the fact that, in the specialized literature, very little data were found about the densities of the composites used, they were measured using the techniques described as follows. For each composite material used, cylinders of approximately 30 mm in length and 10 mm in diameter were made. Each cylinder was created whit successive layers of 2 mm, which were compacted and photopolymerized in turn, for 20 s, according to the indications given by the manufacturers.

To determine the masses, the ELB 300 electronic scale was used, and the masses were determined for the six cylinders made of composite materials. To precisely determine the volumes of the six cylinders, they were three-dimensionally scanned, one at a time, using a 3DSystems Capture 3D scanner. A number of 12 successive scans were performed. In the next phase, the 12 scans were “united” into a single surface, then procedures were applied to remove the so-called “artifacts”, then non-conforming surfaces were removed, and the gaps were “filled”. The model, composed of a perfectly closed surface, was exported to the SolidWorks program, where it was automatically transformed into a virtual solid. Using the Tools/Evaluate/Mass Properties command, the volume of the scanned cylinder was measured and centralized.

Knowing that: ρ = m/V (1), where: ρ—density of a material; m—mass; V—the volume. Applying this formula, the densities of the six composite materials were determined and centralized in Table 3.

To scan the molar with the prepared cavity, respectively, with each layer of composite applied to restore the morphology, a Systems Capture 3D scanner was used to obtain a virtual model. A 3M Elipar Deepcure-L LED light curing lamp was used for light curing composite materials. For each layer of composite applied, a light curing of 20 s according to the manufacturer’s instructions was performed [53,54,55,56,57,58,59]. As specific software, the following were used: Geomagic (3D Systems, Rock Hill, South Carolina, USA) is a program that allows the processing of “point cloud” geometries obtained either by three-dimensional scanning or after CT scanning and is specific to Reverse Engineering. In the study, only three-dimensional scanning was used. SolidWorks (Dassault Systèmes, Velizy-Villacoublay, France) is a Computer-Aided Design (CAD) software for Direct Engineering. A model obtained in Geomagic and containing perfectly closed surfaces is automatically transformed in SolidWorks into virtual solids [60].

To simulate the mechanical behavior, Ansys Workbench (Ansys, Inc., Canonsburg, Pennsylvania, USA) software was used; it is a software that operates with specific Finite Element Analysis (FEA) techniques. The following methods were used in this study: dental cavity restoration methods and techniques (snow plow, injection-molded and Bichacho), Direct Engineering methods (Direct Engineering), in particular CAD-specific methods, Reverse Engineering methods, which allowed three-dimensional scanning and primary processing of geometric models, The Finite Elements Method (FEM) which is based on the division of virtual solids into smaller volumes, to the nodes of which linear equations are applied that represent the approximation of differential equations describing a physical phenomenon [61].

Analyzing the three cavity reshaping techniques, it was found that, geometrically, the snow plow and injection-molded techniques produce the same three-dimensional pattern. The model obtained by the Bichacho technique differs from that obtained by the other two techniques in that the first layer of fluid composite is 1 mm thick instead of 0.5 mm. It was also identified that a correct model, reflecting cavity restoration, shows the following configuration: a layer of fluid composite (0.5 mm for the snow plow and Injection Molded techniques and 1 mm for the Bichacho technique) applied as a base to both the vertical and horizontal components, proximal wall made of composite paste, at the occlusal level, a layer of paste composite, obtained by combining the four cusps, reworked one by one, to obtain the morphology of the occlusal face.

First, a three-dimensional scanning of the tooth was performed, and a class II cavity was prepared. In order to scan, the molar was fixed in a silicone vascular holder. Initially, 12 successive scans were taken and merged into a single surface, as shown in Figure 1.

The operation was repeated after each composite layer, and virtual models of the composite layers were obtained by volume subtraction operations. These are shown in Figure 2.

Finally, the virtual molar model was obtained where the occlusal mesial cavity was restored with composite materials by the snow plow or injection-molded methods, as shown in Figure 3a–c. Dentin and pulp models were also modeled in Geomagic using CAD and reverse engineering techniques, as shown in Figure 3.

To determine the geometric model of the restored tooth using the Bichacho technique, a similar procedure was followed, and the model shown in Figure 4 was obtained.

In order to analyze the mechanical behavior of a molar model using the snow plow or Injection molded technique and the Admira Fusion and Admira Fusion Flow composite materials, applying bruxism-specific loads, the following procedure was used: the model defined in the SolidWorks Assembly module has been loaded into Ansys Workbench. In the first phase, the model was split into tetrahedron finite elements. This resulted in 2,048,479 nodes and 1,206,256 elements, and this structure is shown in Figure 5.

In the Engineering Data module, the mechanical properties of the materials in the analysis (Young’s modulus, Poisson’s ratio) were entered, established by studying the specialized literature [33,34,35,36,37,38,39,40,41,42,43,44,45,46,47,48,49,50,51,52], and also the densities determined in the study.

A mechanical constraint was introduced, i.e., the molar was considered fixed in the root zone, as shown in Figure 6 (in shades of green).

The position and location of masticatory forces in the cusp area were determined, as shown in Figure 7 (shaded red) [62,63,64].

The duration of the action of the forces was considered to be t = 10 s, and their value would increase from 100 N to 800 N [65,66,67]; the force variation graph is shown in Figure 8.

In order to analyze the mechanical behavior of a molar model using the snow plow or injection-molded technique and Grandio and Grandio Flow composite materials, applying bruxism-specific loads, the following procedure was followed: for this simulation, the following were retained: finite element structure, mechanical constraints (molar root fixation), position, direction, size and duration of loads. The engineering Data module has been updated where the mechanical properties of the materials in the analysis have been introduced using the properties of the Grandio composite materials. Proceeding as above, simulations were performed for the other methods and material combinations, applying bruxism-specific loads.

## 3. Results

Maps showing strains, displacements and von Mises stresses were obtained for the materials and methods used. Figure 9, Figure 10 and Figure 11 show the displacement, strain and stress maps for the snow plow and injection-molded technique and the Admira Fusion and Admira Fusion Flow composite materials.

Figure 12, Figure 13 and Figure 14 show displacement, strain and von Mises stress maps for snow plow or injection-molded and Grandio and Grandio Flow composite materials.

Figure 15, Figure 16 and Figure 17 show displacement, strain and von Mises stress maps for snow plow or Injection Molded and Filtek Supreme composite materials.

Figure 18, Figure 19 and Figure 20 show displacement, strain and von Mises stress maps for the Bichacho technique and Admira Fusion and Admira Fusion Flow composite materials, applying bruxism-specific loads.

Figure 21, Figure 22 and Figure 23 show displacement, strain and von Mises stress maps for the Bichacho technique (classical technique) and Grandio and Grandio Flow composite materials.

Figure 24, Figure 25 and Figure 26 show displacement, strain and von Mises stress maps for the Bichacho technique (classical technique) and composite materials such as Filtek Supreme XT and Filtek Supreme Flow.

By analyzing the result maps where the maximum values were highlighted, the comparative diagrams in Figure 27, Figure 28 and Figure 29 were obtained. Analyzing the results in terms of displacements and strains, it can be seen that the maximum values were recorded in the Bichacho technique and the Filtek Supreme XT and Filtek Supreme flow composite materials. The minimum values were recorded for the snow plow and injection-molded techniques and the Admira fusion and Admira fusion flow composite combination.

Analyzing the displacement and strain diagrams, it can be seen that for each material combination (Filtek supreme and Filtek supreme flow, Grandio and Grandio flow, Admira fusion and Admira fusion flow), in general, the best results were obtained with the Bichacho technique. Thus, it can be considered that as the thickness of the fluid composite increases, so does the elasticity of the restoration. Although there were differences between the strain and displacement values, they were relatively small, which validates all three methods and the materials used. Thus, the attention is oriented on the strain values and their location.

As for the von Mises stress values of the study undertaken, they were similar for all methods, except for the snow plow and injection-molded techniques using Grandio and Grandio flow composites, where the maximum stress value was double, as shown in Figure 29.

## 4. Discussion

According to the study’s null hypothesis (h0), regardless of the application technique and light curing, all combinations of restorative materials react similarly when high occlusal stresses are applied. The study started from the hypothesis that the mode of stress distribution and the lifetime of the materials used do not vary significantly.

The null hypothesis is rejected by the study findings. These observed changes in the mechanical behavior of the composites have clinical significance.

At the level of the oral cavity, through the contraction of the masticatory muscles, functional and parafunctional forces occur, which cause the appearance of stress at the level of the dental-periodontal complex (teeth, alveolar bone, gingival and periodontal tissue) [68]. Special attention should be paid to the stresses occurring in restored teeth. The determination of the distribution and analysis of these stresses are of great importance, as they can contribute to reducing the risk of failure of dental restorations [69]. Factors influencing the strength of restored teeth include the type of cavity, the amount of lost tooth tissue, the filling technique as well as the composition of the used filling materials [70,71]. Second-class cavities are the most prone to fracture due to the often loss of a large volume of tooth tissue, but mostly due to marginal ridge damage [72,73]. Removal of one marginal ridge results in up to 46% loss of tooth strength, and the loss of two marginal ridges results in a 63% loss of strength [74].

Ideally, the filling material should show similar physicochemical properties to the replaced dental tissues, especially the modulus of elasticity [26]—an ideal that is still difficult to achieve [75]. Although many filling materials currently exist, choosing the best material specific to a given situation can be challenging [73]. In an attempt to overcome some of the shortcomings associated with conventional composite materials, a new type of inorganic-organic hybrid restorative material called ormocer (organically modified ceramic) was developed in 1997 [76]. Paste composite resins are recommended for use in the restoration of posterior teeth due to their improved mechanical properties [77]. In fact, even ormoceramic materials are considered alternatives to amalgam or even fully adequate substitutes. Laboratory studies on ormocer have demonstrated good material performance in terms of polymerization shrinkage [78], wear [79], biocompatibility [80] and marginal integrity [81].

Composite materials, however, have a number of disadvantages, such as polymerization shrinkage and poor adaptation to the cavity wall due to their high vascularity, resulting in microleakage, postoperative sensitivity and air voids with undesirable repercussions on the strength of restorations [82]. Therefore, the problem of using a material with increased fluidity for lining the cavity floor and gingival threshold in class II cavities has been raised [83]. The ability of low-viscosity materials to adapt to dentinal irregularities allows them to create an intimate bond with microstructural cavity defects prior to the placement of the paste-like restorative composite [84]. In our study, the centripetal method as the cavity-filling method was chosen, because studies have shown that fillings obtained in this way have the highest degree of microhardness [85].

Given the benefit of combining two materials with different consistencies, techniques have been introduced to provide improved clinical results: the snow plow technique, injection-molded and the Bichacho technique. The main difference between the listed techniques is the light-curing sequence of the three materials: adhesive system, fluid composite and paste composite. The aim of the development of these techniques was to achieve the tightest possible obturations, especially at the gingival threshold, where marginal microleakage and, consequently, secondary caries and postoperative sensitivity occurs [72].

Since the literature abounds with comparative studies that have analyzed the efficiency of these methods in terms of marginal closure at the gingival threshold, but the performance of these techniques has not been analyzed in terms of mechanical strength [86,87,88,89]. The present study aimed to analyze how these restorative techniques can influence the mechanical properties of Class II restorations. This is of particular importance, considering that in the case of bruxism, forces of up to 800 N occur that may jeopardize the integrity of the restorations [65,66,67,68].

As materials, three topical composite resins were used (Filtek Supreme XT, a nanocomposite, Admira fusion, a normocer and Grandio, a nanohybrid), each of them together with the corresponding fluid composite resin (Filtek Supreme flow, Admira fusion flow, and Grandio flow), and the analysis technique used was the finite element method. The Finite Element Method, a modern numerical stress analysis technique, has the great advantage of being applicable to solid and heterogeneous materials with irregular geometry. It is, therefore, an ideal method for examining the behavior of dental restorations [90]. The method is powerful and adaptable in that it can present detailed information about stress, deformations and displacements in complex structures such as teeth and coronal fillings [91]. Analyzing the three cavity reshaping techniques, it was found that, geometrically, the snow plow and injection-molded techniques produce the same three-dimensional pattern. The pattern obtained by the Bichacho technique differs from that obtained by the two techniques in that the fluid composite layer is 1 mm thick, as opposed to 0.5 mm.

Analyzing the displacement and strain diagrams, it can be seen that for each material combination (Filtek supreme and Filtek supreme flow, Grandio and Grandio flow, Admira fusion and Admira fusion flow), in general, the best results were obtained with the Bichacho technique. Thus, it can be considered that as the thickness of the fluid composite increases, the elasticity of the restoration increases. Although there were differences between the values of deformations and strains, they were relatively small (range from 5.25 × 10^−6^–3.21 × 10^−5^ for displacement, 6.22 × 10^−3^–4.34 × 10^−3^ for strain), which validates all three methods and the used materials. Thus, attention is directed to the strain values and their location. Various studies have investigated the extent to which fluid composites used as liners are able, through their elasticity, to cushion the stresses arising during occlusal stress. This has been analyzed in reference to the thickness of the applied fluid composite layer [92,93,94].

There are doubts about whether a high or low elasticity modulus is preferable for composite restorations. Asmussen et al. studied the influence of the modulus of elasticity of composite materials in class I and II cavity restorations on the stresses generated by occlusal loading. They concluded that occlusal restorations made of composite materials should have a high modulus of elasticity to reduce the risk of marginal damage [95]. Pietro Ausiello et al. concluded after their study that the application of low modulus luting and restorative materials partially absorb deformations under loading and limit the stress intensity [9]. Eliguzeloglu et al. suggested that flexible materials, such as glass ionomer cement, flowable composites or nano-filled adhesives, could reduce the stress under paste composites [96]. According to previous studies, it has been suggested that materials used as liners, which exhibit a low modulus of elasticity, decrease the stresses occurring at the composite resin-cavity interface [97,98]. As far as the von Mises stress values of the study were concerned, they were similar for all methods (250–300 MPa), except for the snow plow/injection-molded technique using Grandio and Grandio flow composites, where the maximum stress value was more than double (above 700 MPa).

In general, from the analysis of the stress diagram, it was found that except for the above-mentioned situation, in the case of all materials, the values were somewhat lower than in the other two techniques in the Bichacho technique. An explanation for this phenomenon can be given by the greater thickness of the fluid composite layer in the case of the Bichacho technique, which is more elastic, and can act as a stress absorber. However, the differences recorded between the techniques were small, which is why we cannot consider that the thickness of the fluid composite layer acted as a stress absorber. This has also been found in other studies, such as the one by Min-Kwan Jung. In this study, lithium disilicate, lithium trisilicate, glass ionomer cement, and fluid composite, in thicknesses of 0.5 and 1 mm, respectively, were used as base materials, and the extent to which the type and thickness of the base material can influence the stress distribution was investigated. The results showed that neither the type of base material nor the thickness significantly influenced the stress distribution. However, the absence of a base filling resulted in significantly increased stress [99].

Based on previous research, a general opinion prevails that a material’s behavior as a stress absorber cannot be predicted using the modulus of elasticity alone since polymerization kinetics is a complex material-specific phenomenon [100,101]. Therefore, controversy is still present in the literature regarding the supposed positive effect of flowable composites on stress reduction and marginal integrity. There are studies that indicate that the cavity design and the restorative used material influence the quality of adhesion as well as the stress distribution [102,103].

Regarding the distribution of maximum stress, studies show that in the case of an uninjured tooth, stress is transmitted uniformly along the tooth structures, from enamel to dentin, without critical concentrations being observed at stresses up to 600 N [99,100]. When analyzing the stress maps in this study, it can be seen that the maximum stress was located at the junction between composite and dental enamel, i.e., at the junction of the composite layers. The stress indicates the areas where the material may fail, especially if the stress is cyclic, as in bruxism. Similar to the study undertaken, Pietro Ausiello et al. also found through the finite element method that the maximum stresses in the case of obturation of a mesio-occluso-distal cavity with a bulk composite were also located at the level of the junction between the filling and the cavity walls [102]. The same location of the maximum stresses was found by Aline A. Bicalho et al. in a study carried out using the finite element method on restorations located on the posterior area [103]. Ulla Pallesen observed in her study that 70% of coronary obturation fractures occurred in patients with parafunctions [104].

The maximum stress shows that these locations can be explained by the existence of sharp edges and peaks in these areas, which in mechanics are considered to be stress concentrators. This has negative repercussions on the restored tooth, as the fracture zones are critical areas where carious damage can occur.

It is important to mention that besides the limitation given by the use of the finite element method (the materials were considered to be homogeneous, isotropic and linearly elastic), the results of the study could not be compared with similar studies. Considering that no studies were found in the specialized literature to analyze the mechanical behavior (but mainly on the marginal microleakage) of the combination of these three obturation techniques and the selected materials or similar, it was not possible to obtain validation through a comparison with a similar study. Therefore, in the discussions, the results of the study were only compared with the results of other studies in terms of the location of the maximum stresses recorded, respectively, and the influence of the fluid composite layer in damping the von Mises stresses.

## 5. Conclusions

Finite element analysis showed that the fillings made by combining paste composite and composite flow variants in the case of Admira Fusion and Filtek Suprem XT are more stable in terms of maximum von Mises stress compared to the combination of Grandio and Grandio Flow and, therefore, more resistant to stress.

The fact that the maximum displacements and strains did not show large differences between the snow plow and injection-molded methods, regardless of the used materials, proves that these material combinations have similar elasticity, and the method of application in the class II cavity does not influence their properties.

The finite element study carried out for the Bichacho technique showed higher displacements and strains compared to the other two dental restoration techniques (snow plow and injection-molded), which supports the observation that as the thickness of the fluid composite increases, the elasticity of the restoration increases.

Another finding revealed that the thickness of the fluid composite layer applied as an intermediate layer does not play an important role in stress relief. The only exception is the use of Grandio and Grandio flow composites. In the case of the application of Grandio flowable composite in a thin layer of 0.5 mm, double von Mises stresses were recorded compared to the application of a 1 mm layer.

With the exception of the use of Grandio and Grandio flow composites with the snow plow and injection-molded techniques, all other techniques and material combinations provided similar results, with the selection of the technique and the used materials being left to the choice of the practitioner.

## Figures and Tables

**Figure 1 jfb-14-00354-f001:**
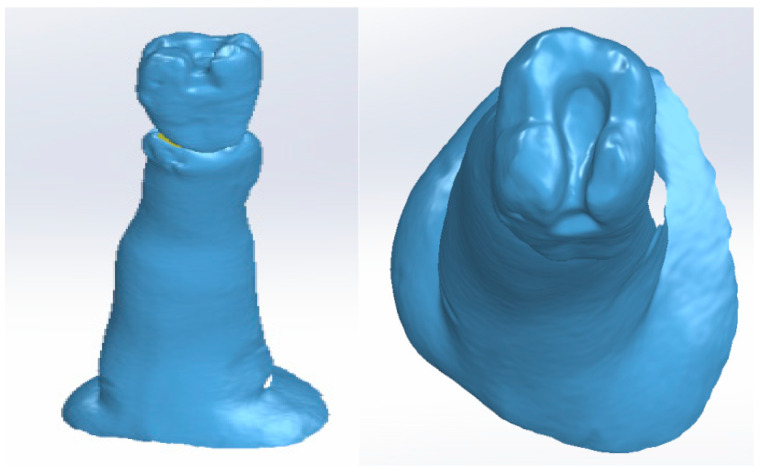
The unified surface of the successive scans.

**Figure 2 jfb-14-00354-f002:**
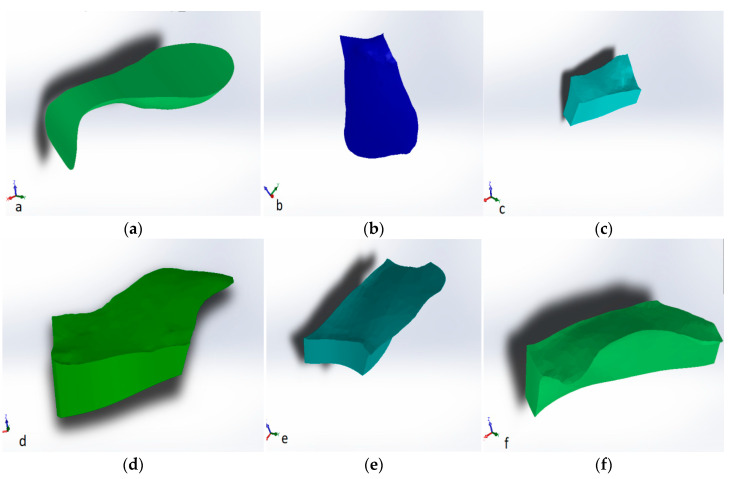
Virtual models of composite layers: (**a**) fluid composite layer applied to the pulp and parapulp wall; (**b**) proximal wall made of paste composite; (**c**) paste composite restoration of the inner slope of the mesio-vestibular cusp; (**d**) paste composite restoration of the inner slope of the disto-vestibular cusp; (**e**) paste composite restoration of the inner slope of the disto-lingual cusp; and (**f**) paste composite restoration of the inner slope of the mesio-lingual cusp.

**Figure 3 jfb-14-00354-f003:**
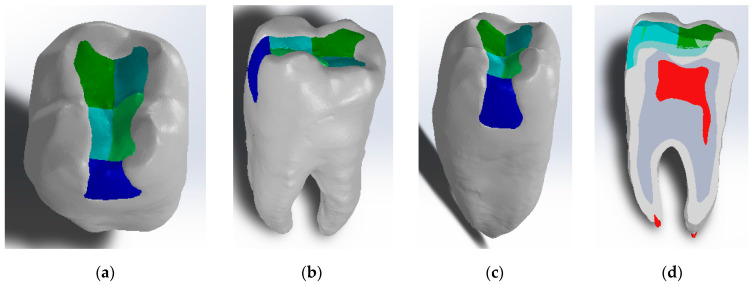
Virtual molar model after the application of composite layers by the snow plow and injection-molded method: (**a**) occlusal view; (**b**) buccal view; (**c**) mesial view; and (**d**) planar section through the model.

**Figure 4 jfb-14-00354-f004:**
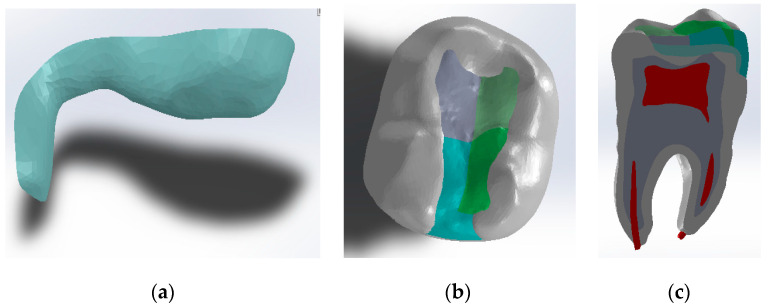
Geometric model of the molar using the Bichacho technique: (**a**) Model of the fluid composite layer of about 1 mm; (**b**) Top view of the model; (**c**) Planar section through the model.

**Figure 5 jfb-14-00354-f005:**
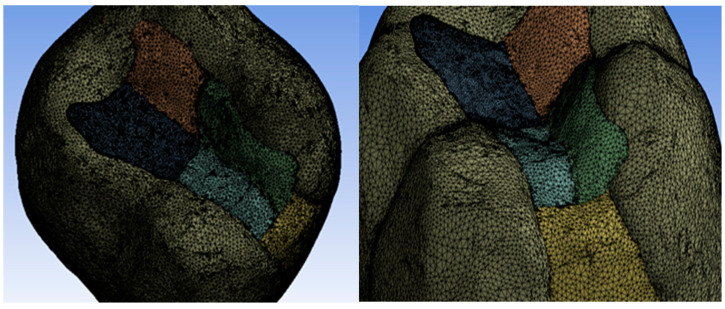
Finite element structure of the model.

**Figure 6 jfb-14-00354-f006:**
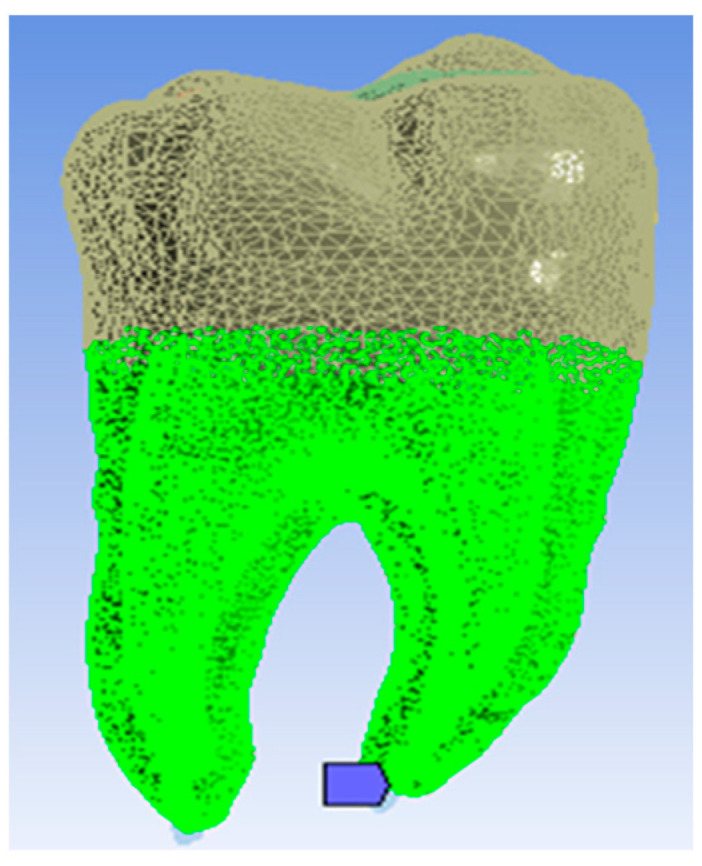
The areas considered fixed.

**Figure 7 jfb-14-00354-f007:**
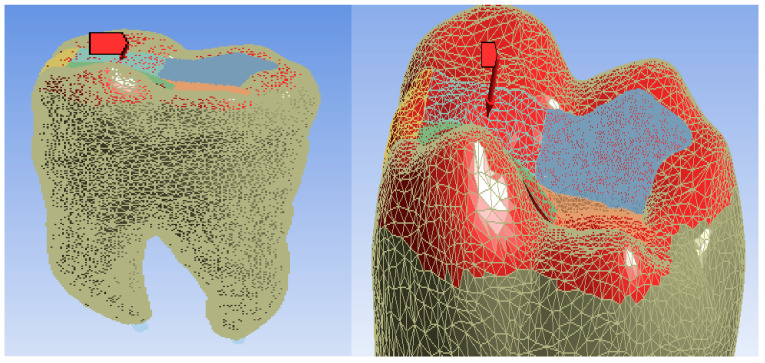
Position and location of forces.

**Figure 8 jfb-14-00354-f008:**
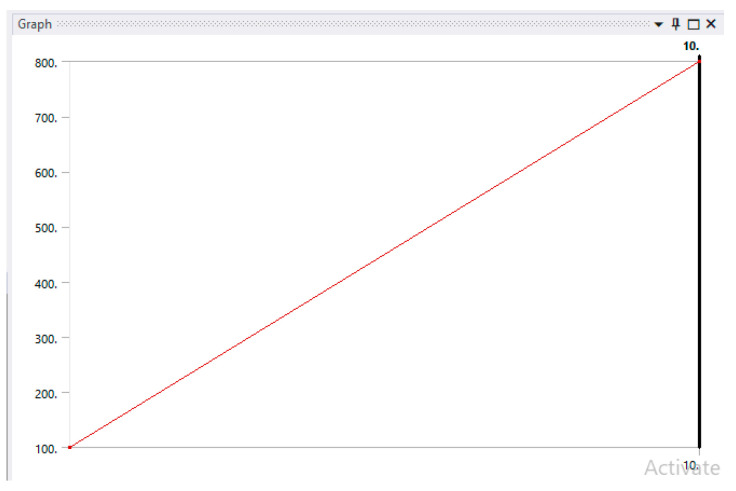
Temporal variation of force from 100 to 800 N.

**Figure 9 jfb-14-00354-f009:**
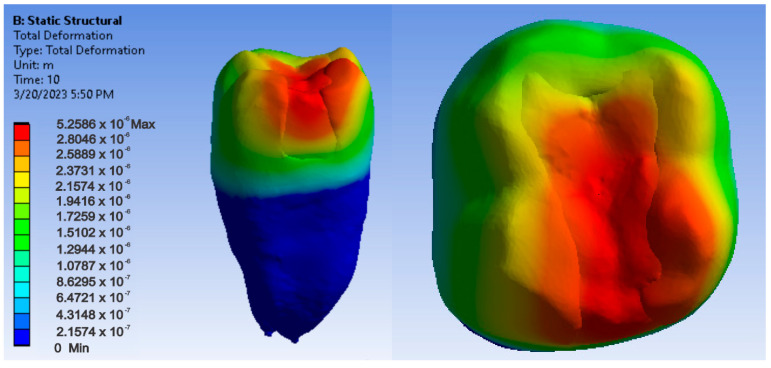
Admira Fusion and Admira Fusion Flow displacement map by the snow plow and injection-molded technique.

**Figure 10 jfb-14-00354-f010:**
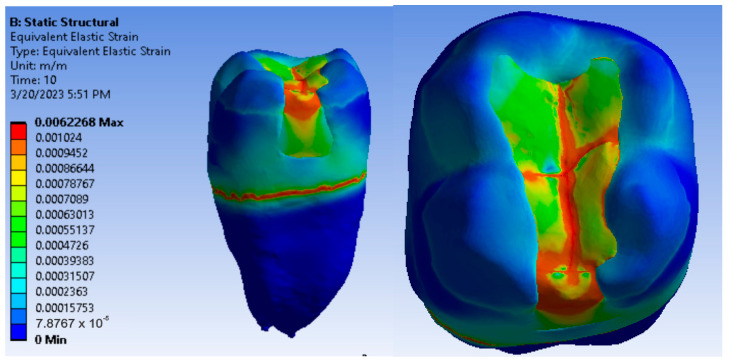
Strain Map Admira Fusion and Admira Fusion Flow by the snow plow and injection-molded technique.

**Figure 11 jfb-14-00354-f011:**
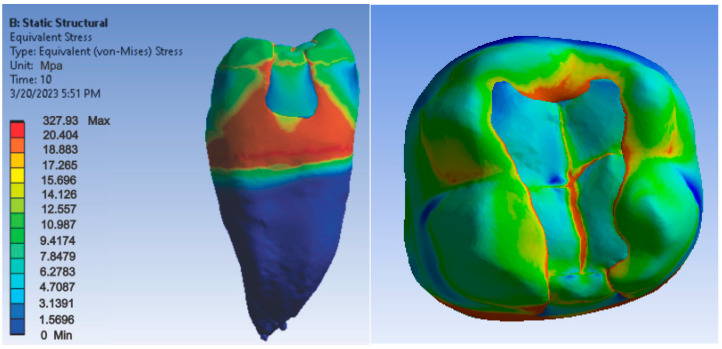
Admira Fusion and Admira Fusion Flow von Mises stress map by the snow plow and injection-molded technique.

**Figure 12 jfb-14-00354-f012:**
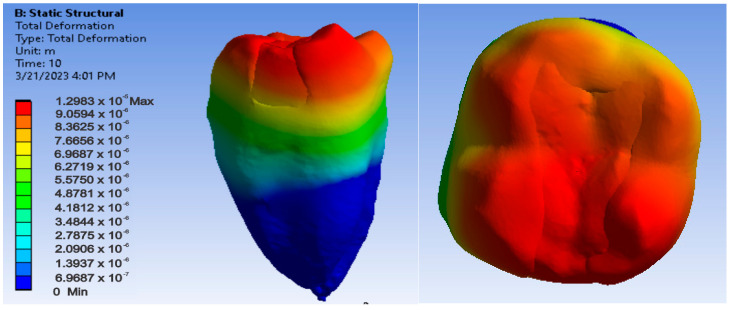
Grandio and Grandio Flow displacement map by the snow plow and injection-molded techniques.

**Figure 13 jfb-14-00354-f013:**
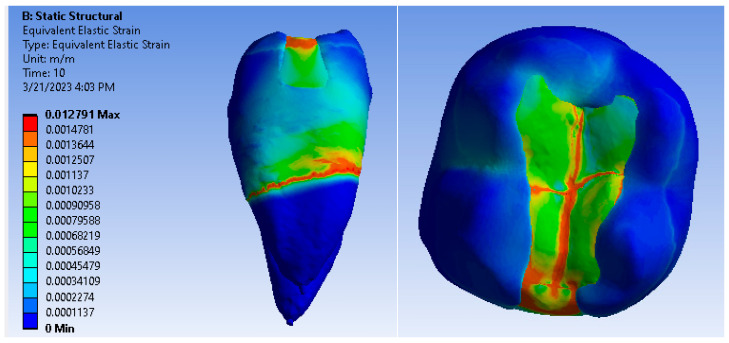
Grandio and Grandio Flow strain map by the snow plow and injection-molded technique.

**Figure 14 jfb-14-00354-f014:**
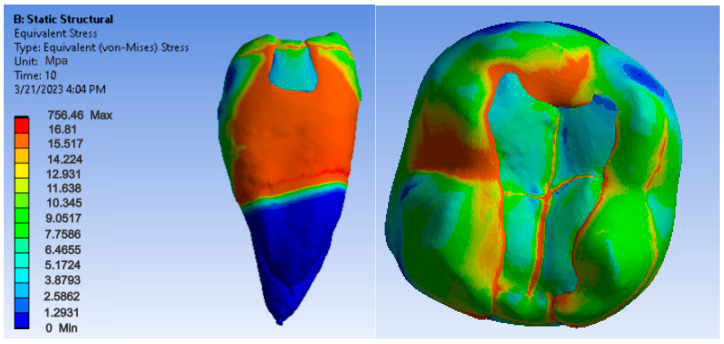
Grandio and Grandio Flow von Mises stress map by the snow plow and Injection molded techniques.

**Figure 15 jfb-14-00354-f015:**
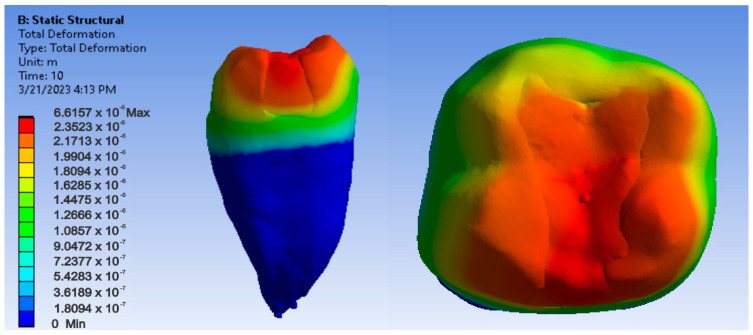
Displacement map of Filtek Supreme XT and Filtek Supreme Flow by the snow plow and injection-molded techniques.

**Figure 16 jfb-14-00354-f016:**
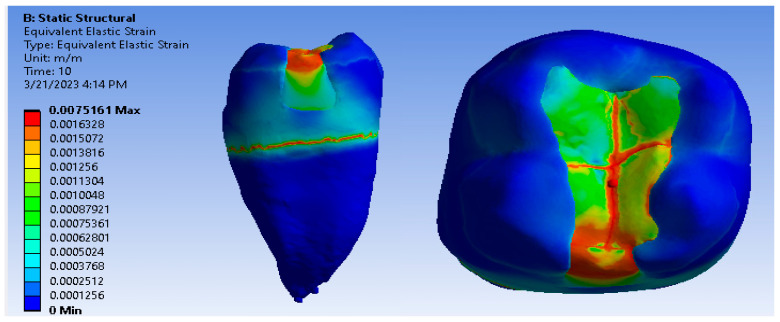
Strain map of Filtek Supreme XT and Filtek Supreme Flow by the snow plow and injection-molded techniques.

**Figure 17 jfb-14-00354-f017:**
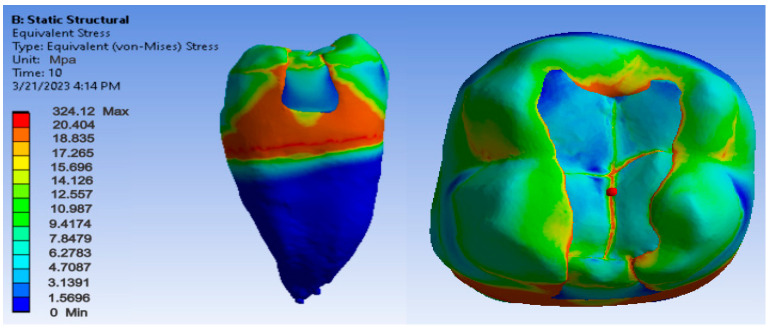
Filtek Supreme XT and Filtek Supreme Flow von Mises stress map with the snow plow and injection-molded techniques.

**Figure 18 jfb-14-00354-f018:**
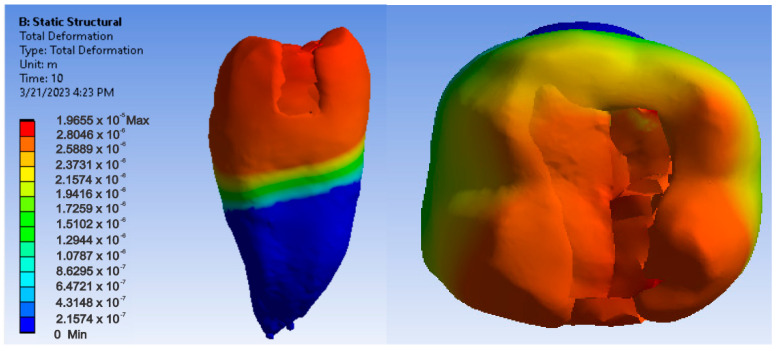
Admira Fusion and Admira Fusion Flow displacement map using the Bichacho technique.

**Figure 19 jfb-14-00354-f019:**
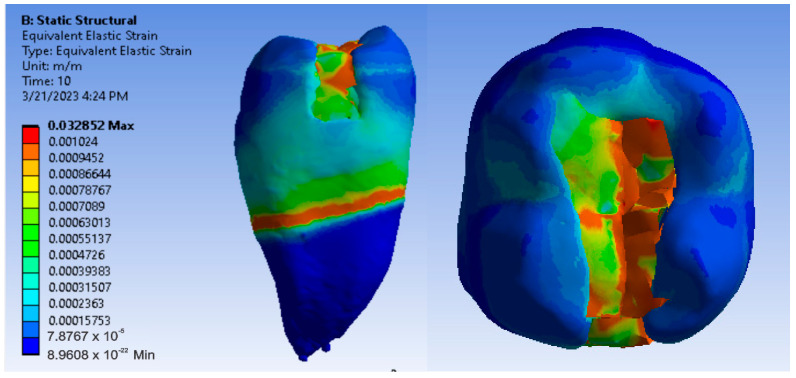
Strain map of Admira Fusion and Admira Fusion Flow by the Bichacho technique.

**Figure 20 jfb-14-00354-f020:**
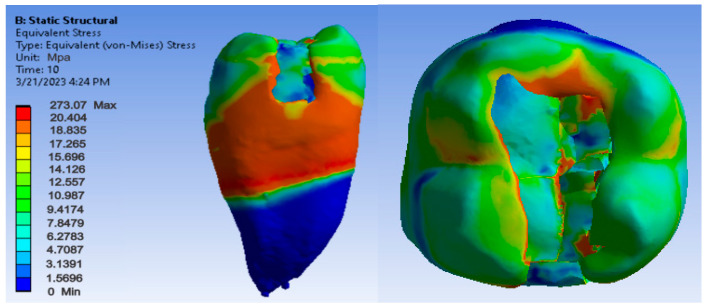
Admira Fusion and Admira Fusion Flow von Mises stress map by the Bichacho technique.

**Figure 21 jfb-14-00354-f021:**
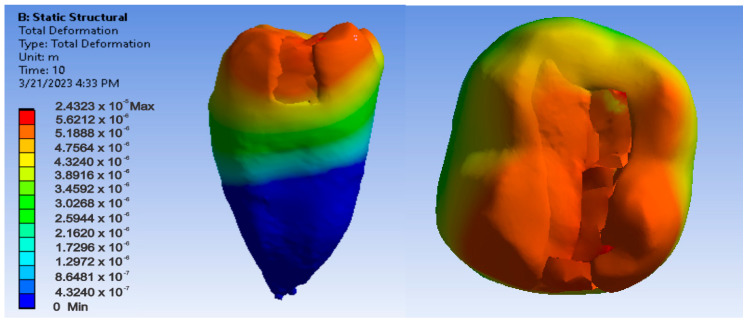
Map of the Grandio and Grandio Flow displacements by the Bichacho technique.

**Figure 22 jfb-14-00354-f022:**
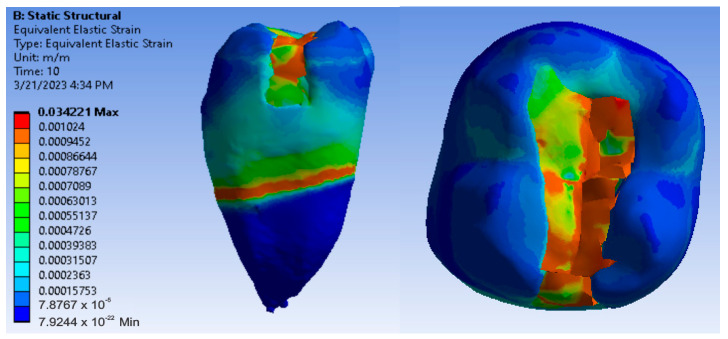
Grandio and Grandio Flow strain map by the Bichacho technique.

**Figure 23 jfb-14-00354-f023:**
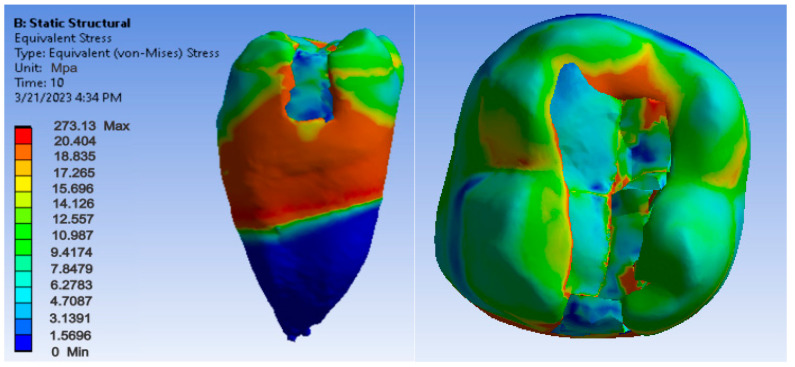
Map of Grandio and Grandio Flow von Mises stress by the Bichacho technique.

**Figure 24 jfb-14-00354-f024:**
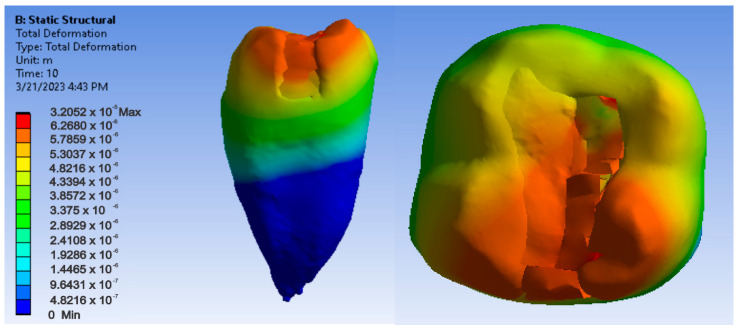
Displacement map of Filtek Supreme XT and Filtek Supreme Flow by the Bichacho technique.

**Figure 25 jfb-14-00354-f025:**
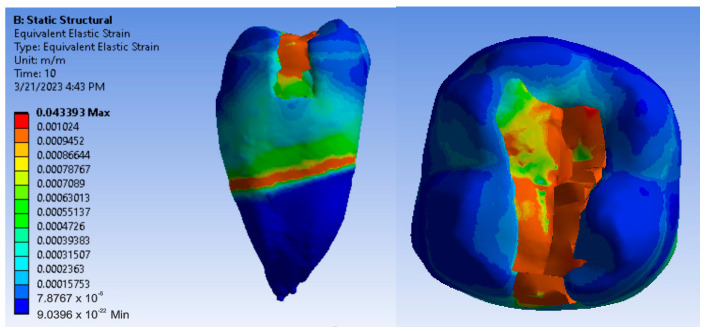
Strain map of Filtek Supreme XT and Filtek Supreme Flow by the Bichacho technique.

**Figure 26 jfb-14-00354-f026:**
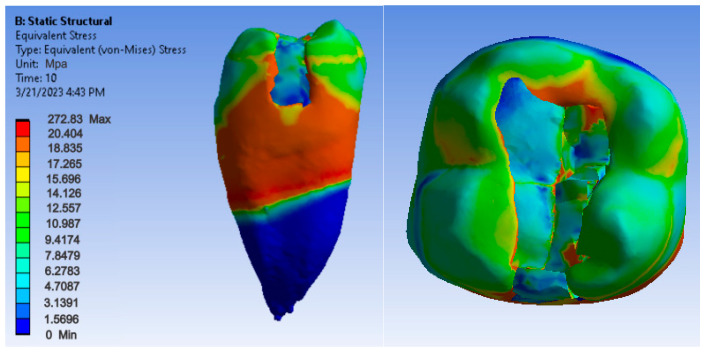
Filtek Supreme XT and Filtek Supreme Flow von Mises stress map by the Bichacho technique.

**Figure 27 jfb-14-00354-f027:**
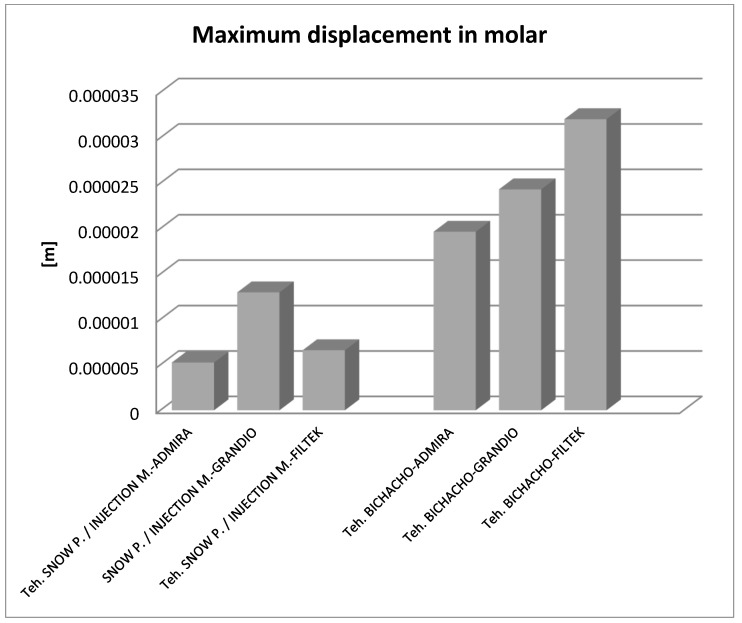
Comparative diagram of maximum displacements.

**Figure 28 jfb-14-00354-f028:**
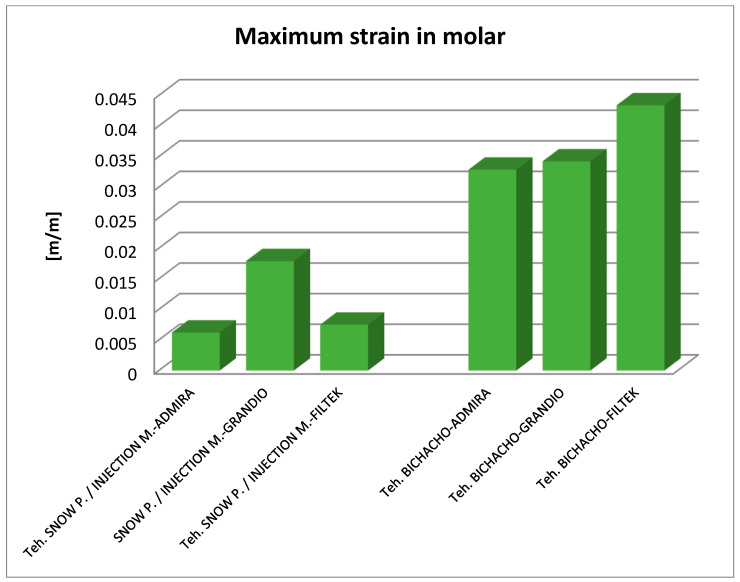
Comparative diagram of maximum strains.

**Figure 29 jfb-14-00354-f029:**
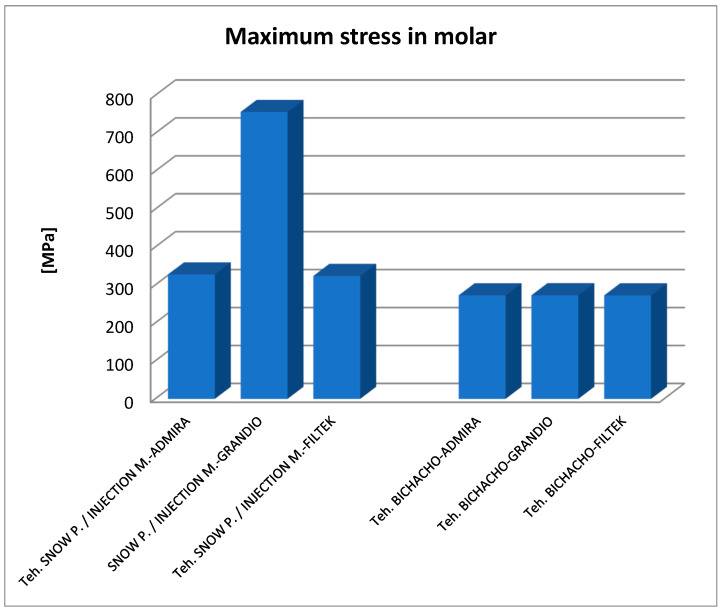
Comparative diagram of maximum von Mises stress.

**Table 1 jfb-14-00354-t001:** The composite materials are used together with their chemical composition.

Name of Composite	Type of Composite	Matrix	Filler Content/Filler Size (By Volume)	Manufacturer
Grandio Flow	Universal nano-hybrid	Bis-GMA, TEGDMA and HEDMA	Glass-ceramic, nanoparticle, 65.6%	VOCO, Cuxha ven, Germany
Admira Fusion Flow	Nano-hybrid ORMOCER	Silicon oxide	Silicon oxide, 74%	VOCO, Cuxha ven, Germany
Filtek Supreme Flow	Nanocomposite	Bis-GMA, TEGDMA and Procrylat resins	Silica nanofiller, zirconia nanofiller and zirconia/silica nanocluster 55.5%	3M Oral Care
Grandio	Universal nano-hybrid	Bis-GMA, UDMA, TEGDMA	Mixture of different dimethacrylates, silicate fillers, initiators, pigments, amines, additives, 87%	VOCO, Cuxha ven, Germany
Admira Fusion	Nano-hybrid ORMOCER	Silicon oxide	Silicon oxide, glass-ceramic filler 60%	VOCO, Cuxha ven, Germany ceramic (Ormocer)
Filtek Supreme XT	Nanocomposite	Bis-GMA, UDMA, TEGDMA, Bis-EMA resins	Non-agglomerated/non-aggregated 4 to 11 nm zirconia filler and aggregated zirconia/silica cluster filler, 63.3%	3M Oral Care

Bis-GMA—Bisphenol A-Glycidyl Methacrylate; TEGDMA—Tri-ethylene Glycol Dimethacrylate; HEDMA—hexamethylene dimethacrylate; UDMA—Urethane Dimethacrylate; Bis-EMA—Bisphenol A-Ethoxylated Dimethacrylate.

**Table 2 jfb-14-00354-t002:** Physico-mechanical properties of the used composite materials.

Component	Density [Kg/m^3^]	Young Elasticity Modulus [Pa]	Poisson Coefficient
Enamel	2958	7.79 × 10^10^	0.3
Dentine	2140	1.76 × 10^10^	0.25
Pulp	1100	1.75 × 10^9^	0.4
Admira fluid composite layer	2107.9	3.29 × 10^9^	0.34
Admira paste composite layers	1931.2	9.8 × 10^9^	0.31
Grandio Fluid Composite Layer	2180.1	6.85 × 10^9^	0.31
Grandio paste composite layers	2097.9	7.9 × 10^9^	0.31
Filtek Supreme fluid composite layer	2080.9	7.8 × 10^9^	0.393
Filtek Supreme paste composite layers	1853.7	5.76 × 10^9^	0.45

**Table 3 jfb-14-00354-t003:** Densities of composite materials.

Composite	Density [Kg/m^3^]
ADMIRA fluid	2107.9
GRANDIO fluid	2180.1
FILTEK fluid	2080.9
ADMIRA paste	1931.2
GRANDIO paste	2097.9
FILTEK paste	1853.7

## Data Availability

The authors declare that the data from this research are available from the corresponding authors upon reasonable request.

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
