# Peer review of "Study on the Restoration of Class II Carious Cavities by Virtual Methods: Simulation of Mechanical Behavior"

_jfb, 2023, doi:10.3390/jfb14070354_

Round 1

Reviewer 1 Report

All other combinations of techniques and materials have had similar results, so the choice of technique and materials is left to the doctor. Some comments given as follows:

1.      Line 55, type or restoration needs to discussed for better understanding.

2.      Line 68, just Bruxism? Several challenges needs to explained and then give the rationalisation why bruxism is the crucial one.

3.      Line 93, please make it more sound science with using passive form, do not use “we”.

4.      Line 113, the thick of 0.5 mm is the standard or what? Give the explanation.

5.      Line 119, for the explanation in table 1 needs to make more comprehensive, the present one is too simple.

6.      What is the novel bought by the authors in the current submission? Its works have been widely discussed in the past. Nothing something really new in the present form. The lack of a novel seems to make the present submission like to replication/modified work. The authors need to detail their novelty in the introduction section. It is a major concern for rejecting this paper.

7.      The authors needs to explain in the introduction section related the urgency of performing computational simulation, why not clinical and/or experimental study? It bring several advantages compared both of them, such as lower cost and faster results. Please explain this information along with relevant reference as follows: https://doi.org/10.3390/ma14247554, and https://doi.org/10.3390/biomedicines11030951

8.      How the authors determine the number of element used in the computational simulation model? It is based from mesh convergence study? If yes, please explain it. And if not, the present results is not appropriate and should be rejected. It aims to select computational models with an optimal number of elements, which does not use too many elements in order to not burden the computational load, whilst still being able to provide accurate results. Please explain this information along with relevant reference as follows: https://doi.org/10.3390/ma16093298

-

Author Response

21 June 2023

Dear Reviewer

It is an immense pleasure to submit our revised manuscript entitled: “Study on the Restoration of Class II Carious Cavities by Virtual Methods. Simulation of Mechanical Behavior”.

We are grateful for the attention and effort spent in reviewing our work, and valuable comments made by the respectful editor and reviewers.

We sincerely hope that the revised manuscript is now suitable for publication in the Journal of Functional Biomaterials.

The authors ensured that the manuscript meets Journal of Functional Biomaterials style requirements and have made changes to the manuscript according to the suggestions from the editor and the reviewers.

Find answers to your comments below.

1. Line 55, type or restoration needs to discussed for better understanding.

The type of restoration and its characteristics have been reformulated and developed within the introduction.

2. Line 68, just Bruxism? Several challenges needs to explained and then give the rationalisation why bruxism is the crucial one.

Considering the specifics of the article, which aims to analyze the behavior of class 2 restorations under mechanical stress, bruxism was the most relevant considering that within this parafunction, much greater forces are developed than physiological ones. For a better understanding of this aspect, I have reformulated and completed the paragraph.

3. Line 93, please make it more sound science with using passive form, do not use “we”.

Passive form was used to replace the “we” form.

4. Line 113, the thick of 0.5 mm is the standard or what? Give the explanation.

This thickness of 0.5 mm is not standardized by any method. We considered it necessary to apply the fluid composite at the level of the horizontal component in the same thickness as for the vertical component in order to highlight the influence of the thickness of the fluid composite layer in the mechanical behavior of the restorations.

5. Line 119, for the explanation in table 1 needs to make more comprehensive, the present one is too simple.

Additional explanations were added for the correct framing of table number 1.

6. What is the novel bought by the authors in the current submission? Its works have been widely discussed in the past. Nothing something really new in the present form. The lack of a novel seems to make the present submission like to replication/modified work. The authors need to detail their novelty in the introduction section. It is a major concern for rejecting this paper.

Although such studies have been addressed in the past (comparison of the 3 centripet techniques: Snow plow, Injection molded and Bichacho), the novelty elements brought into the study are given by the multitude of materials analyzed, in correlation with a series of specific factors: the important dimension of cavities, respectively the high forces at which the restorations were simulated. We also consider it very important to continue research in this direction, considering the appearance of new composite materials and the maintenance of increased frequency of proximal dental cavities.

7. The authors needs to explain in the introduction section related the urgency of performing computational simulation, why not clinical and/or experimental study? It bring several advantages compared both of them, such as lower cost and faster results. Please explain this information along with relevant reference as follows: https://doi.org/10.3390/ma14247554, and https://doi.org/10.3390/biomedicines11030951

The clinical and/or experimental study does not allow the determination of tensions, displacements and deformations inside the tooth.
Also the advantages of using FEM were completed and highlighted in the introduction, with the insertion of the suggested references.

8. How the authors determine the number of element used in the computational simulation model? It is based from mesh convergence study? If yes, please explain it. And if not, the present results is not appropriate and should be rejected. It aims to select computational models with an optimal number of elements, which does not use too many elements in order to not burden the computational load, whilst still being able to provide accurate results. Please explain this information along with relevant reference as follows: https://doi.org/10.3390/ma16093298

The number of finite elements was based on the mesh convergence study and it was determined automatically, using the default settings of the program. Even if the number of finite elements is relatively large, the calculation time for the analyzed situations was a few minutes for each simulation. Suggested reference was added.

Reviewer 2 Report

The article “Study on the Restoration of Class II Carious Cavities by Virtual Methods. Simulation of Mechanical Behavior.” aimed to analyze the mechanical behavior of several topical composite materials (nanocomposites, nano-hybrids and ormocer) using different application techniques using an FEA model.

The article requires major revisions.

There is no null hypothesis. Please provide it in the introduction and accept or reject it in the discussion.

Even if not mandatory for an MDPI journal, the reviewer thinks that a structured abstract could be beneficial (Materials and methods, results, conclusion).

Rather than Bichacho, the reviewer suggests using “centripetal approach”. 

Lines 85-6:

The authors should cite one or more studies using FEA to study and analyze mechanical behavior. FEA is used for direct restorations (already cited: 66. Ausiello, P.; Ciaramella, S.; Di Rienzo, A.; Lanzotti, A.; Ventre, M.; Watts, D.C. Adhesive class I restorations in sound molar 632 teeth incorporating combined resin-composite and glass ionomer materials: CAD-FE modeling and analysis. Dent. Mater. 2019, 633 35, 1514–1522. [PubMed] ), for endo-treated teeth (suggested reference: https://doi.org/10.1080/01694243.2017.1304172) or for prosthetic appliances (already cited: 6. Ausiello, P.; Rengo, S.; Davidson, C.L.; Watts, D.C. Stress Distributions in Adhesively Cemented Ceramic and Resin-Composite 510 Class II Inlay Restorations: A 3D-FEA Study. Dent. Mater. 2004, 20, 862–872. )

From the first part of the Materials and Methods, it seems that “filling first the vertical cavity on the mesial side and then the horizontal cavity on the occlusal side. “ has been planned. This can be true for the centripetal approach (the ones the authors call Bichacho). But this can’t be performed clinically with the snowplow technique of injection molding (unless to have several silicone indexes which are not clinically convenient or easy to manage. 

In fact, in line 106 the authors write: “but each layer is not light-cured separately, but all three layers simultaneously ”, so how is it possible to correctly manage the centripetal approach?

The authors should also add filler characteristics in Table 1.

Line 332: what is Odontally???

Lines 345-7: these lines can be safely removed

Line 449-451:

Please support this concept with a similar study with the same findings on stress concentration. The reviewer suggests: https://doi.org/10.1080/01694243.2017.1304172

minor spell issues.

Author Response

21 June 2023

Dear Reviewer

It is an immense pleasure to submit our revised manuscript entitled: “Study on the Restoration of Class II Carious Cavities by Virtual Methods. Simulation of Mechanical Behavior”.

We are grateful for the attention and effort spent in reviewing our work, and valuable comments made by the respectful editor and reviewers.

We sincerely hope that the revised manuscript is now suitable for publication in the Journal of Functional Biomaterials.

The authors ensured that the manuscript meets Journal of Functional Biomaterials style requirements and have made changes to the manuscript according to the suggestions from the editor and the reviewers.

Find answers to your comments below.

  1. There is no null hypothesis. Please provide it in the introduction and accept or reject it in the discussion.

According to the study's null hypothesis (h0), regardless of the application technique and light curing, all combinations of restorative materials react similary when high occlusal stresses are applied.

The null hypothesis is rejected by the study findings. These observed changes in the mechanical behavior of the composites have clinical significance. I made the changes in the article.

  1. Even if not mandatory for an MDPI journal, the reviewer thinks that a structured abstract could be beneficial (Materials and methods, results, conclusion).

The article was written considering the request from the specific template of this special issue, namely We strongly encourage authors to use the following style of structured abstracts, but without headings

  1. Rather than Bichacho, the reviewer suggests using “centripetal approach”. 

The use of the term Bichacho was preferred, considering that the other two techniques (Snow plow and Injection molded) are also centripetal restoration methods. (References 79)

  1. Lines 85-6:

The authors should cite one or more studies using FEA to study and analyze mechanical behavior. FEA is used for direct restorations (already cited: 66. Ausiello, P.; Ciaramella, S.; Di Rienzo, A.; Lanzotti, A.; Ventre, M.; Watts, D.C. Adhesive class I restorations in sound molar 632 teeth incorporating combined resin-composite and glass ionomer materials: CAD-FE modeling and analysis. Dent. Mater. 2019, 633 35, 1514–1522. [PubMed] ), for endo-treated teeth (suggested reference: https://doi.org/10.1080/01694243.2017.1304172) or for prosthetic appliances (already cited: 6. Ausiello, P.; Rengo, S.; Davidson, C.L.; Watts, D.C. Stress Distributions in Adhesively Cemented Ceramic and Resin-Composite 510 Class II Inlay Restorations: A 3D-FEA Study. Dent. Mater. 2004, 20, 862–872. )

Content from suggested references have been inserted into the text.

  1. From the first part of the Materials and Methods, it seems that “filling first the vertical cavity on the mesial side and then the horizontal cavity on the occlusal side. “ has been planned. This can be true for the centripetal approach (the ones the authors call Bichacho). But this can’t be performed clinically with the snowplow technique of injection molding (unless to have several silicone indexes which are not clinically convenient or easy to manage. 

In fact, in line 106 the authors write: “but each layer is not light-cured separately, but all three layers simultaneously ”, so how is it possible to correctly manage the centripetal approach?

Within the text, the stages specific to each technique were better highlighted.

Thus, in the Snow plow technique, the adhesive is applied and photopolymerized. A layer of fluid composite 0.5 mm thick is applied over it. Paste composite is applied over the non-polymerized fluid composite layer, and the 2 layers are polymerized simultaneously.

In the Injection molded technique, the 3 layers: the adhesive, the fluid composite (thickness 0.5 mm) and the paste composite are applied in turn, but photopolymerized simultaneously.

In the Bichacho technique, the 3 materials: the adhesive, the fluid composite (thickness of 1 mm) and the paste composite are applied in turn, the photopolymerization being carried out separately, for each of the applied materials.

  1. The authors should also add filler characteristics in Table 1.

The requested component was added in table 1.

  1. Line 332: what is Odontally???

The term was used incorrectly and implicitly deleted from the text.

  1. Lines 345-7: these lines can be safely removed

The paragraph has been deleted from the text.

  1. Line 449-451:

Please support this concept with a similar study with the same findings on stress concentration. The reviewer suggests: https://doi.org/10.1080/01694243.2017.1304172

Thank you for the suggested reference. This has been added to the text, along with other relevant references.

Reviewer 3 Report

Line 99: snow plow technique is repeated.

line 129: "I studied the literature" is not a form of sentence adequate for a scientific article. Please reword and expand this section since you cite 21 articles after a 4 word sentence.

line 214 and 215, as in line 129 this part needs to be reworked and expanded. 

you did not perform any statistical analysis of the data: why?

Minor English language editing is needed. Check the text and read it aloud to see if it's fluid or not. Some sentences are very long and exhausting to read.

Author Response

21 June 2023

Dear Reviewer

It is an immense pleasure to submit our revised manuscript entitled: “Study on the Restoration of Class II Carious Cavities by Virtual Methods. Simulation of Mechanical Behavior”.

We are grateful for the attention and effort spent in reviewing our work, and valuable comments made by the respectful editor and reviewers.

We sincerely hope that the revised manuscript is now suitable for publication in the Journal of Functional Biomaterials.

The authors ensured that the manuscript meets Journal of Functional Biomaterials style requirements and have made changes to the manuscript according to the suggestions from the editor and the reviewers.

Find answers to your comments below.

  1. Line 99: snow plow technique is repeated.

The repetition of the term has been deleted.

  1. line 129: "I studied the literature" is not a form of sentence adequate for a scientific article. Please reword and expand this section since you cite 21 articles after a 4 word sentence.

The mentioned phrase has been reformulated in the passive form.

The multitude of references is justified by the fact that, in order to carry out the necessary simulations, it was necessary to centralize the density, Young's modulus and Poisson's modulus for all 6 composite materials used.

The lack of articles that centrally address all these aspects, led to the need to mention all the articles that presented the necessary information.

  1. line 214 and 215, as in line 129 this part needs to be reworked and expanded. 

The method of obtaining the data was additionally clarified, as well as a detailed presentation of the protocol in which the densities of the composite materials used were obtained.

  1. You did not perform any statistical analysis of the data: why?

Thanks for the suggestion. The simulations in the study were performed on a single tooth (lower 2nd molar), for which the physico-mechanical properties (Young's modulus, Poisson ratio, density) were selected from the specialized literature, remaining the same for all simulations.

The same principle was applied in the case of the 6 composites, which led to obtaining a single result for each individual simulation.

Therefore, performing a statistical analysis was not necessary.

  1. Minor English language editing is needed. Check the text and read it aloud to see if it's fluid or not. Some sentences are very long and exhausting to read.

Changes have been made to improve for a more fluent reading of the content.

Round 2

Reviewer 1 Report

Well effort form authors in the previous revision. Some correction still needed as follows.

1.      The abstract section should include quantitative results.

2.      Please end your abstract with a "take-home" message.

3.      Reorder keywords based on alphabetical order.

4.      Line 121, regarding materials assumption in computational simulation via finite element method, the present article does not explain it. The reviewer suggest it is assumed as homogeneous, isotropic, and linear elastic as common assumption in computational simulation. For giving better understanding, please provide this information and support with relevant reference as follows: https://doi.org/10.3390/su142013413, https://jurnaltribologi.mytribos.org/v33/JT-33-31-38.pdf, and https://doi.org/10.3390/jfb12020038

5.      Line 198, The simulated model using computed tomography (CT) scan model. Please explain in brief the basic concept and application of CT in medical purpose. Also, refer the relevant reference as follows: https://doi.org/10.1016/j.heliyon.2022.e12050, https://doi.org/10.3390/su15010823, and https://doi.org/10.3390/biomedicines11020427

6.      Since the present study performing computational simulation, the present results need to verified with identical study from analytical, experimental, and/or computational results. However, the present study does not provide this validation process. Please state it as limitation of the present study and refer some previous relevant literature as follows: https://doi.org/10.1038/s41598-023-30725-6, https://doi.org/10.3390/fluids7070225, https://doi.org/10.3390/jfb13020064, https://doi.org/10.1080/23311916.2023.2218691, and https://doi.org/10.3390/met12081241

7.      The authors need to improve the discussion in the present article to become more comprehensive. The present form was insufficient.

-

Reviewer 2 Report

All comments have been amended

Author Response

We are grateful for the attention and effort spent in reviewing our work, and valuable comments made by the respectful editor and reviewers.